# The Biomarker Potential of miRNAs in Myotonic Dystrophy Type I

**DOI:** 10.3390/jcm9123939

**Published:** 2020-12-04

**Authors:** Emma Koehorst, Alfonsina Ballester-Lopez, Virginia Arechavala-Gomeza, Alicia Martínez-Piñeiro, Gisela Nogales-Gadea

**Affiliations:** 1Neuromuscular and Neuropediatric Research Group, Institut d’Investigació en Ciències de la Salut Germans Trias i Pujol, Campus Can Ruti, Universitat Autònoma de Barcelona, 08916 Badalona, Spain; ekoehorst@igtp.cat (E.K.); aballester@igtp.cat (A.B.-L.); amartinezp.germanstrias@gencat.cat (A.M.-P.); 2Centre for Biomedical Network Research on Rare Diseases (CIBERER), Instituto de Salud Carlos III, 28029 Madrid, Spain; 3Neuromuscular Disorders Group, Biocruces Bizkaia Health Research Institute, 48903 Barakaldo, Spain; virginia.arechavalagomeza@osakidetza.eus; 4Ikerbasque, Basque Foundation for Science, 48009 Bilbao, Spain; 5Neuromuscular Pathology Unit, Neurology Service, Neuroscience Department, Hospital Universitari Germans Trias i Pujol, 08916 Badalona, Spain

**Keywords:** myotonic dystrophies, miRNAs, biomarkers, therapeutics

## Abstract

MicroRNAs (miRNAs) are mostly known for their gene regulation properties, but they also play an important role in intercellular signaling. This means that they can be found in bodily fluids, giving them excellent biomarker potential. Myotonic Dystrophy type I (DM1) is the most frequent autosomal dominant muscle dystrophy in adults, with an estimated prevalence of 1:8000. DM1 symptoms include muscle weakness, myotonia, respiratory failure, cardiac conduction defects, cataracts, and endocrine disturbances. Patients display heterogeneity in both age of onset and disease manifestation. No treatment or cure currently exists for DM1, which shows the necessity for a biomarker that can predict disease progression, providing the opportunity to implement preventative measures before symptoms arise. In the past two decades, extensive research has been conducted in the miRNA expression profiles of DM1 patients and their biomarker potential. Here we review the current state of the field with a tissue-specific focus, given the multi-systemic nature of DM1 and the intracellular signaling role of miRNAs.

## 1. Introduction

MicroRNAs (miRNAs) are small, single-stranded RNAs about 22 nucleotides long that regulate gene expression by either inhibiting translation or promoting degradation of their target mRNAs [1]. It is estimated that the human genome encodes over a thousand miRNAs, which can target dozens of mRNAs, and every individual mRNA can be targeted by several miRNAs [2]. miRNAs are estimated to regulate about one-third of human protein-coding genes [3]. The majority of miRNAs result from RNA polymerase II transcription, which yields long primary miRNA (pri-miRNA) transcripts. Pri-miRNAs are trimmed in the nucleus by the RNase Drosha, yielding premature hair-looped miRNA of ± 70 nucleotides. These pre-mRNAs are transported to the cytoplasm where they are further processed by the RNase Dicer, resulting in mature miRNA. Mature miRNAs are then incorporated into the RNA-induced silencing complex (RISC), where the miRNA strand anneals to the 3’ untranslated regions (UTRs) of target mRNAs, leading to the degradation or translation inhibition of mRNAs and subsequent protein repression [3]. miRNAs are mostly known for their gene regulation properties. However, they also play an important role in intercellular signaling and can therefore be found abundantly in bodily fluids, including blood and urine [4]. There are several ways in which miRNAs can reach the bloodstream from the tissues, including lipid or lipoprotein complexes and extracellular vesicles (EVs), but they have also been found as free-floating complexes with Argonaute proteins [5]. This makes them excellent biomarker candidates, since one of the most important properties of a good biomarker is easy accessibility. Furthermore, their detection is easily achieved with low-cost techniques, such as quantitative reverse transcription PCR. In addition, miRNA expression profiles are different in the diseased state and in their tissue-specific expression patterns, further strengthening their biomarker potential, and extensive research has been conducted on this discovery. The most studied field is oncology, where miRNAs encapsulated in exosomes are suggested to function as biomarkers in the diagnosis and prognosis of several cancers, including breast cancer, prostate cancer, and colorectal cancer [6]. In addition, miRNAs have been found to play an essential role as biomarkers in numerous other diseases, such as Alzheimer’s disease [7], epilepsy [8], sepsis [9], and in neurodegenerative and neuromuscular disorders [10,11].

Myotonic Dystrophy type I (DM1) is an autosomal dominant muscle dystrophy, with a multi-systemic nature and an estimated prevalence of 1:8000 [12]. Patients display a core set of symptoms, which include muscle weakness, myotonia, respiratory failure, cardiac conduction defects, cataracts, and endocrine disturbances. The DM1 phenotype can be divided into five clinical categories, each with distinct clinical features: congenital, childhood-onset, juvenile-onset, adult-onset, and late-onset DM1. In congenital DM1, symptoms arise at birth or during the first month of life and include hypotonia, respiratory failure, feeding difficulties, failure to thrive, and clubfoot deformities. Mortality is especially high in the perinatal period due to respiratory failure [13]. The diagnosis of childhood-onset DM1 is often missed due to uncharacteristic symptoms. Affected children show cognitive and learning abnormalities after the first year. None of the ‘classic’ DM1 symptoms, such as muscle myopathy and myotonia, is present at first, but they often develop in adulthood [14]. Juvenile DM1 has an onset between 10 and 20 years. It spans the continuum between childhood and adult-onset and has some overlapping features with both types. Nevertheless, they differ from childhood-onset DM1 by the more pronounced muscle involvement and from the adult-onset in the often underdiagnosed cognitive impairment and earlier cardiac and respiratory problems [15]. The classic/adult-onset is the most prevalent DM1 phenotype and arises typically around the second or third decade of life. Core features are progressive muscle weakness and myotonia with preferential involvement of the cranial, trunk, and distal limb muscles and early-onset cataracts [16,17]. In addition, cardiac conduction defects are common in adult-onset DM1, which is a major contributor to increased mortality and sudden death in patients [18]. Late-onset or mild DM1 patients usually start to show symptoms after the age of forty. Symptoms include mild muscle weakness, premature cataract, and balding. Not only is the disease heterogeneous in disease onset, within the different subtypes much heterogeneity exists as well in symptom development. Symptom manifestation can range from mild muscle weakness with myotonia to loss of ambulation with cardiac conduction defects and severe cataracts. At the point of diagnosis, it is unclear which subset of symptoms will develop over time.

DM1 is caused by a Cytosine-Thymine-Guanine (CTG) expansion in the in the 3’ untranslated region of the myotonic dystrophy protein kinase gene (*DMPK*). Unaffected individuals carry 5–35 CTG repeats, whereas the length of the CTG expansion in patients can range from fifty to thousands of CTGs and has been associated with age of symptom onset and disease severity [19,20]. DM1 is considered an RNA gain-of-function disorder, in which expanded transcripts accumulate as intranuclear RNA foci. These foci can sequester muscleblind-like (MBNL) proteins, which subsequently lead to diminished activity and downstream deregulation of the alternative splicing of several genes [21,22]. In addition, the RNA foci cause hyper-phosphorylation and thus upregulation of the CUG-BP and ETR-3-like factors family member 1 (CELF1) via several signaling kinases [22,23]. CELF1 is also a splice factor, and its inappropriate phosphorylation results in deregulation of downstream target genes. Disruption of the alternative splicing of several genes can be directly correlated to symptoms arising in DM1. For example, mis-splicing of the ClC-1 chloride channel leads to reduced chloride conductance in muscle fibers, which is known to produce myotonia [24], and alternative splicing of cardiac troponin T and the insulin receptor is correlated with cardiac defects and diabetes, respectively [21]. However, these spliceopathies cannot explain the full multi-systemic picture found in DM1, and the discoveries of bidirectional transcription of the *DMPK* gene, aberrant DNA methylation, repeat-associated non-ATG translation, and miRNA deregulation have added to the vast complexity of DM1 pathology [25,26,27,28].

In DM1, the need for a biomarker that can predict disease outcomes is essential, especially for cardiac and respiratory complications, to improve patient survival. Symptoms and disease progression are highly variable among patients, and life expectancy is significantly decreased, primarily due to respiratory failure and cardiac events [29]. This high variability and the multi-systemic nature of the disease present a unique challenge in terms of disease management. Currently, disease management is mostly focused on symptomatic treatment and preserving function and independence. Current assessment strategies include genetic testing, electromyography, skeletal muscle histopathology, and magnetic resonance. The only circulating marker used to date is creatine kinase (CK), but this is not elevated in all patients and lacks disease specificity, since it is altered in most disorders involving skeletal muscle damage [30], as well as intense physical activity [31]. Other suggested biomarkers are the alternative splicing changes observed in the DM1 skeletal muscle [32]. However, monitoring them would require several invasive muscle biopsies. miRNAs have the potential to help estimate the degree and type of involvement of symptoms before they appear as well as to apply the appropriate preventive measures. For example, several studies have indicated that miRNAs correlate to muscle strength and disease stage in DM1 [33,34]. In addition to disease management, miRNAs could be used as biomarkers for future clinical trials. Currently, outcome measures consist mostly of clinical and functional ones, such as cognitive function assessment by questionnaires, patient-reported outcomes such as DM1-activ, muscle testing by the 6 min walk test, grip strength, Muscular Impairment Rating Scale (MIRS), and myotonia degree [35]. These can be, however, influenced by several confounding factors. For example, the 6 min walk test is not only influenced by muscle strength but also by factors such as the level of fatigue and motivation [36]. These tests also might lack sensitivity to subtle changes happening in the rather short time frame of most clinical trials.

Here we review the current knowledge on the potential of miRNAs as a biomarker for disease development in DM1. Due to the multi-systemic nature of DM1 and the different tissues involved, the advances will be discussed per tissue, and we will touch on the therapeutic potential of miRNAs in DM1.

## 2. miRNAs in Skeletal Muscle

Several endogenous miRNAs are ubiquitously expressed, such as let-7, but a subset of miRNAs is only expressed in certain tissues. An example is the muscle-specific miRNAs (myo-miRs), which are expressed in skeletal muscle and cardiac muscle. They consist of miR-1, miR-133a/b, and miR-206, which are highly enriched in skeletal muscle, whereas miR-1 and miR-133a/b are also highly enriched in cardiac muscle. miR-1 plays an important role in skeletal muscle growth and differentiation by regulating the serum response factor (SRF) and myocyte enhancer factor-2 (MEF2) [37]. Both activate skeletal muscle gene expression in collaboration with myogenic basic helix loop helix proteins. miR-133 was initially thought to drive proliferation by repressing the expression of SRF, indicating an opposing role from miR-1 [38]. However, recent reports suggest a similar regulating function to miR-1, promoting differentiation in a similar way [39]. Both miR-1 and miR-133 are also involved in the modulation of electrical conductance in the heart [40]. miR-206, on the other hand, is solely expressed in skeletal muscle and promotes myoblast differentiation [41]. Later on, miR-208a/b, miR-499, and miR-486 were added to the list of myo-miRs. These new members are striated muscle-specific, and miR-208a is found only in cardiac tissue. Both miR-208 and miR-499 are encoded in slow-twitch, type I myosin genes and are required to establish the slow-twitch fiber phenotype [42]. miR-486 is potentially involved in muscle growth and homeostasis by regulating the Phosphoinositide 3-kinase/Protein kinase B signaling pathway [43]. Myo-miRs have been suggested to play a role in several diseases, including cardiac hypertrophy, cardiac arrhythmias, and muscular disorders [44,45,46,47].

Since DM1 is a muscle dystrophy first, and therefore the most affected tissue is the muscle, naturally the first studies on miRNA dysregulation as a potential biomarker for disease were conducted in the muscle (Figure 1, Table 1). It was first studied by Gambardella et al. in 2010 [48]. They focused on myo-miRs, in addition to miR-103 and miR-107, which were proposed to be attractive candidates for binding *DMPK* mRNA [49]. More specifically, a computational analysis revealed that miR-103 and miR-107 would repress the *DMPK* wild-type allele by 15%, and that repression of mRNA by CTG repeat-binding miRNAs would increase with the number of CTGs in the 3′UTR. They proposed a model in which CTG repeat-binding miRNAs, such as miR107 and miR-103, preferentially bind to mutated *DMPK* mRNA, which could have a miRNA-leaching effect and, therefore, implications for DM1 pathology [49]. Only miR-206 was found to be overexpressed in biopsies of DM1 patients. However, the predicted target genes of miR-206, such as Utrophin, were not altered. A potential explanation can be the presence of multiple myofibers in skeletal muscle. For example, miR-206 is known to be highly enriched in regenerating fibers [50], which constitutes only a small portion of the bulk tissue measured. Although miR-206 shows a pronounced increase due to the addition of regenerating fibers, this does not mean that its target genes are equally affected, and minor changes might not be detected in bulk tissue. Interestingly, miR-206 was localized in centralized nuclei and nuclear clumps in DM1 muscle sections, both hallmarks of DM1 histopathology. In the following year, Perbellini et al. [51] conducted a similar study in muscle biopsies from 15 DM1 patients, but with very different results. They broadened their study to 24 miRNA candidates, which have important regulatory roles in skeletal muscle or were found to be dysregulated in Duchenne muscular dystrophy. They found miR-1 and miR-335 to be upregulated and miR-29b/c and miR-33 to be downregulated. Predicted targets involved in muscle development (MEF2a, MET, GATA6, and HAND2), arrhythmia (KCNE1, KCNJ2, CALM2), splicing (SFRS9), and atrophy (DAG1, DIABLO, RET, TRIM63, TGFB3) of miR-1 and miR-29 were significantly upregulated in DM1 patients, indicating functional relevance and a possible contribution to DM1 pathology. In addition, miR-1 was vastly present in the internal nuclei and in nuclear clumps of DM1 myofibers, similar to the miR-206 distribution found by Gambardella et al. [48]. Although expression levels were found not to be dysregulated, the cellular distribution of miR-206 and miR-133b was severely altered in patients compared to controls, in a similar way as miR-1.

The paper by Fernandez-Costa et al. [52] adopted a very different approach to studying miRNA dysregulation of DM1 by using a *Drosophila* model, in which CTG expansions were introduced. They found 20 miRNAs to be differentially expressed, of which 19 were downregulated and 1 was upregulated. Only three of those were preserved in humans, miR-1, miR-7, and miR-10, which underwent a validation study in muscle biopsies from five DM1 patients. The downregulation was preserved in humans, and at least seven of their target genes were upregulated. For miR-1, these were the antioxidant enzyme superoxide dismutase 1 (SOD1), the transcriptional regulator SWI/SNF-related, matrix-associated, actin-dependent regulator of chromatin, subfamily A, member 4 (SMARCA4), and the nucleotide exchange factor Neuroepithelial Cell Transforming 1 (NET1). miR-7-upregulated targets included amyloid precursor protein secretase Cathepsin B, autophagy regulator cysteine protease (ATG4), and cytoskeletal protein Vinculin (VCL), and miR-10 caused an upregulation of Ubiquitin-activating enzyme E1 (UBE11), a member of the small ubiquitin-like modifier (SUMO)ylation machinery. This suggests that miRNA dysregulation in DM1 is involved in a wide range of pathological mechanisms. The finding of miR-1 downregulation is in contrast to the results found by Perbellini et al. [51], since they found miR-1 to be upregulated. To further elucidate the mechanism involved in the dysregulation of these miRNAs, they performed several experiments on the pri-miRNA precursors of these miRNAs and found that expanded CTG repeats decreased the levels of the primary precursor of miR-7 in *Drosophila* flies, and this was also observed in skeletal muscle biopsies of DM1 patients. In addition, they found that Muscleblind, the homolog of human MBNL1, is necessary for the regulation of miR-1 and miR-7 in *Drosophila* flies. MBNL1 protein in humans is trapped by RNA foci in DM1 and has been previously described to participate in the biogenesis of miR-1 [1]. Although it will have to be validated in humans, their results in flies suggest that the depletion of MBNL1 in DM1 can be an explanation for the miR-1 downregulation in skeletal muscle biopsies of DM1 patients. They also made the first link to the physiological relevance of miRNA dysregulation, albeit only in flies. Overexpression of miR-10 in the DM1 drosophila model increased their lifespan.

The vast difference in results between the studies, which at first glance look very similar, might be explained by the use of different muscle biopsies. The first study was conducted in biopsy material from the vastus lateralis, whereas the second study used material coming from the biceps brachii, and the last study used a mixture of material from the vastus lateralis, biceps brachii, and deltoid. Since in DM1, in general, the observed muscle weakness is prominently distal, and only in later stages of the diseases are the proximal muscles, such as the bicep brachii and the vastus lateralis, involved, it could be that the stage of each individual patient at the moment of biopsy can influence the obtained results. In addition, miRNA expression levels are studied by quantitative PCR (qPCR), where expression levels are normalized against an endogenous miRNA of which the levels are known to be stable or by using a spike-in miRNA. Total miRNA content has been shown to be an experimental variable and adds a systematic bias in miRNA quantification [54]. No gold standard for this type of normalization exists, and groups often use different approaches, decreasing the comparability of their studies. In the case of above studies, all three studies opted for a different normalization approach, a potential explanation for their contradictory results. However, Fritegotto et al. [53] recently published a study in which they adopted a very similar experimental design to Gambardella et al. [48]. They used the same muscle biopsy material and the same housekeeping gene for normalization, but their results match only in part. miR-206 is upregulated, which is in consensus with what Gambardella et al. [48] found in 2010. However, they found additionally that miR-1 and miR-133a/b were significantly decreased. This result is in consensus with the findings of Fernandez-Costa [52], but in vast contrast to the study by Perbellini et al. [51], since they found miR-1 to be upregulated. A possible explanation can be that Perbellini et al. [51] used control subjects that were admitted for suspected neuromuscular disorders. This can skew their results since these ‘control’ subjects may have an underlying muscular disorder. Interestingly, this study by Fritegotto et al. [53] was the first study to try to correlate their miRNA dysregulation findings to DM1 pathology. A histopathological score was assigned to each muscle biopsy. However, the histopathological score and disease duration did not correlate significantly to miRNA expression levels. They did observe that the lowest levels of miR-1 and miR133a/b and the highest levels of miR-206 were found in patients with the most severe histopathological score or longest disease duration. Thus, even though Fritegotto et al. [53] and Gambardella et al. [48] adopted the same experimental procedure, their results differ. It might be due to the use of different clinical subtypes of DM1. In Fritegotto et al.’s study [53], the clinical subtypes ranged from childhood to adult onset. Unfortunately, Gambardella et al. [48] only disclosed the current age range and not the age of onset, so the clinical subtype is unknown, but it might be relevant. Childhood-onset DM1 patients display a different array of symptoms compared to adult-onset DM1, suggesting that different pathological mechanisms are at play.

Although studies in skeletal muscle have given us valuable information on miRNA dysregulation in DM1, it can be argued that it does not have the best biomarker potential since obtaining a muscle biopsy is an invasive procedure, especially if you consider that the diagnosis of DM1 can be done with a simple genetic test using a blood sample, which makes muscle biopsies in DM1 scarce and, most of the time, unnecessary. The studies did give us important information on potential targets for therapy development, which will be discussed further on in the review.

## 3. miRNAs in the Heart

Information on miRNA patterns in the heart of DM1 patients is scarce, most likely due to the difficulty of sample collection. To date, two papers have looked into the involvement of miRNAs in cardiac pathology in DM1 (Figure 1, Table 2). First of all, Rau et al. [55] showed in 2010 that miR-1 is downregulated in DM1 patients compared to controls, and that downstream targets of miR-1, the Voltage-dependent L-type calcium channel subunit alpha-1C (CACNA1C) and connexin 43 (GJA1), which are responsible for intracardiomyocyte conductance, are upregulated in heart samples of DM1 patients. The loss of miR-1 was due to MBNL1, which binds to the pre-miR-1, blocking the binding site for LIN-28 and subsequently disrupting the processing of pre-miR-1. The second study, performed by Kalsotra et al. [56], screened a DM1 mouse model for over 500 miRNAs and identified 54 differentially expressed miRNAs. To translate these findings to humans, 22 miRNAs were tested in heart tissue of eight DM1 samples and four controls. Twenty were significantly downregulated in human DM1 heart tissue, including two myo-miRs (miR-1 and miR-133a). Pathway analysis of the misregulated miRNAs revealed the loss of function of the myocyte enhancer factor-2 (Mef2) transcriptional network. In addition, several of the affected miRNAs found in the study have been previously described to produce arrhythmias or fibrotic changes. For example, miR-1, which was identified in both studies, is known to regulate gap junction proteins and cardiac channels, and a reduction in its expression greater than 50% may contribute to the conduction defects found in DM1 [55]. Two other downregulated miRNAs worth mentioning are miR-23a/b, of which postnatal upregulation has been shown to result in CELF1 downregulation [57]. Additionally, upon introduction of expanded CUG repeat RNA into a heart-specific and inducible DM1 mouse model, postnatal upregulation of miR-23a/b is reversed, and CELF1 is subsequently upregulated, proposing an additional mechanism for CELF1 upregulation in DM1.

## 4. miRNAs in Blood and Serum

Although the above studies discussing tissues, muscle, and heart have given us valuable information regarding the involvement of miRNAs in DM1 pathology, due to the invasive nature of sample collection, their biomarker potential is limited. The field, therefore, shifted its focus on circulating miRNAs for biomarkers in DM1 (Figure 1, Table 3). Circulating miRNAs diffuse into the bloodstream from several of the affected tissues, making them excellent candidates as biomarkers. In addition, drawing blood is a simple and much less invasive procedure than obtaining a muscle biopsy.

The first study on circulating miRNAs in DM1 was performed by Perfetti et al. in 2014 [33]. They performed a miRNA panel of 381 miRNAs in twenty-four plasma samples of DM1 patients and twenty-six controls. The fourteen miRNAs that showed a significantly different expression level compared to controls were subjected to validation in a bigger cohort (*n* = 36). In this bigger cohort, nine miRNAs were differentially expressed: miR-133a, miR-193b, miR-191, miR-454, miR-574, miR-885-5p, and miR-886-3p were increased, whereas miR-27b levels were decreased. Pooling the nine miRNAs into a DM1-miRNA score resulted in an accurate discrimination between DM1 patients and controls. miR-133a alone was also able to discriminate between the two populations. Interestingly, both the DM1-miRNA score and miR-133a showed a negative correlation with global muscle strength, assessed by the Medical Research Council (MRC) muscle scale, and significant increases in median miR-133a and DM1-miRNA scores were seen in higher MIRS classes. The following year, Koutsoulidou et al. performed a very similar study in the sera of DM1 patients [34]. The biggest difference is they decided to focus solely on myo-miRNAs and studied the expressions of miR-1, miR-133a, miR133b, and miR-206. miR-133a is the only myo-miRNA that was previously found to be differentially expressed [33]. However, in this study all four myo-miRs were found to be significantly increased in blood sera of twenty-three DM1 patients, and myo-miRs were distinguishable between patients and controls with Receiver Operating Characteristic (ROC) curves ranging from 0.94 to 0.97, or when averaged into a single value with a score of 0.98. In contrast to Perfetti et al. [33], they were unable to find any correlation between the miRNA expression levels and DM1 severity. However, when the groups were stratified, based not on their clinical picture and MRC muscle scale but on their progression in muscle wasting, they found that the four miRNAs were increased in progressive DM1 patients compared to stable patients, and that the miRNAs could discriminate between progressive and stable DM1 patients. Progressive vs. stable was based on whether the patients showed a change in the MRC scale in the past two years. This is the first longitudinal correlation made and a first hint at the real biomarker potential of these miRNAs.

Both studies mentioned above repeated their analysis in bigge r cohorts: Perfetti et al. [58] chose to include the three additional miRNAs found by Koutsoulidou et al. [34] into this validation study (*n* = 103). Out of these twelve combined miRNAs, eight were found to be differentially expressed: miR-1, miR-133a/b, miR-206, miR-140-3p, miR-454, and miR-574 were increased, and miR-27b was decreased. This means that four out of nine of the originally found were not validated in this bigger cohort. Notably, the myo-miRs were able to differentiate between patients and controls, but better results were obtained when all the miRNAs were combined into a DM1-miRNA score, when myo-miRs scores were pooled, or when levels of miR-133a and miR-133b were averaged into a miR-133a/b score. This time they were again able to find a significant, negative correlation with muscle strength and a weak, direct correlation with CK values. An interesting point raised was that miR-133b was significantly higher in female DM1 patients compared to DM1 males, but this difference between gender was not found in healthy subjects or in the original study. Of note, they included DM2 patients in this study as well to see whether the findings in DM1 were translatable to DM2, and all except miR-27b were found to be deregulated in thirty DM2 patients. Koutsoulidou et al. focused first on the four myo-miRs previously reported [34] and validated their differential expression in a cohort of 63 DM1 patients. The previously reported increased levels in progressive DM1 patients compared to stable DM1 patients were also validated [59]. In addition to validating their previous results, they also considered the eight additional miRNAs found by Perfetti et al. in plasma [33]. Six out of eight were found to be significantly increased, whereas no differences were found in miR-454 expression, and miR-27b, previously reported to be significantly decreased in DM1 patients, was shown to be significantly increased. A novel finding was the encapsulation of these myo-miRs in EVs: it was reported that the majority of the differentially expressed myo-miRs in DM1 patients could be found in EVs. The levels of the four myo-miRs were significantly higher in EVs isolated from sera of the DM1 patients compared to controls and seem to discriminate between progressive and stable DM1 patients. However, caution should be taken with regards to these results. It has been shown that polymer-based methods to precipitate EVs (such as the Exoquick^TM^ Exosome Precipitation solution used in this study) do not exclusively precipitate EVs, and co-isolation of other molecules, including RNA-Protein complexes, is a possibility [60]. Especially in the case of miRNAs, it has been shown that these kits may preferentially precipitate non-exosomal miRNAs [61]. This means that the miRNAs found in this precipitate may not exclusively originate from EVs or in fact might primarily be non-exosomal.

Although some discrepancies can be found between the studies of the two groups above, they both show a clear miRNA deregulation pattern in the sera or plasma of DM1 patients. This is in vast contrast to another study by Fernandez-Costa et al. [62] in 2016, which was unable to validate six serum miRNAs as DM1 biomarkers. A panel of 175 known serum miRNAs was tested in ten DM1 patients and controls, of which six potential candidates were chosen, but they were unable to validate these candidates by qPCR. The strongest two candidates, miR-21 and miR-130a, were further tested in a bigger cohort of twenty-one DM1 patients, but no differences were found.

Several correlations between miRNA expression levels and DM1 clinical phenotype have been made, and a recent paper has shown how miRNA expression levels can be a biomarker for rehabilitation [63]. After a six-week exercise rehabilitation training, a significant decrease was found in four myo-miRs, namely miR-1, miR-133a/b, and miR-206, in parallel with an improvement in muscle function. Of note, when the four miRNAs were first analyzed for differences between patients and controls, only miR-1 and miR-206 were significantly increased, while miR133a/b were upregulated only in a subset of the patients. The study does, however, show us the potential of myo-miRs in serum as a rehabilitation biomarker, which would be a nice addition to the now used clinical outcomes, which often lack sensitivity and reproducibility and can be influenced by the multi-systemic nature of the disease.

To date, only one study has looked at the serum and muscle miRNA levels simultaneously [64]. Since the myo-miRs in blood are thought to be a representation of the miRNA levels in muscle, one study used a slightly different approach in that they used whole blood instead of plasma or serum. They found the levels of circulating miR-133a, miR-29b, and miR-33a to be increased in DM1 patients. However, in muscle tissue two myo-miRs (miR-1 and miR-133a) and miR29c were found to be downregulated, which indicates that circulating myo-miRs are not reflective of the situation in the muscle. The increase in miR-133a was previously described by several studies [58,59,63], whereas the other two miRNAs are novel findings. Their results in muscle are similar to the findings of two other studies, although the downregulation of miR-29c has not been reported before [52,53].

Similar to the findings in muscle, several studies focusing on miRNA expression levels in blood show no homogeneous findings. Again, differences in sample sizes and normalization methods chosen could be to blame. For example, Perfetti et al. [33,58] decided to use a combination of three different normalizers, Cel-miR-39, miR-17-5p, and miR-106a, whereas Koutsoulidou et al. [34,59] used miR-16 to normalize their expression profiles. The study by Fernadez-Costa et al. [62], which failed to validate any of the previous findings, used the mean Ct value of miRNAs detected for normalization. Both Perfetti et al. [58] and Koutsoulidou et al. [59] decided to validate their results in a bigger cohort, and both found different results, showing the importance and influence of sample size on experiments. Another important difference between the studies is the use of different blood components (i.e., serum, plasma, or whole blood). It has been shown that the miRNA expression levels differ between serum and plasma, for example due to the activation and release of miRNAs from platelets during collection of plasma [65].

Several things have to be taken into account with regards to extracellular miRNAs. Although some miRNAs are tissue-specific, there is still a major overlap between the expression of miRNAs. For example, although miRNA-206 is highly enriched in skeletal muscle, it can also be detected in cardiac muscle, and its origin might be from both. The same holds true for miR-1 and miR-133, which are highly enriched in both tissues. The multi-systemic nature of DM1 may add additional complexity to determining the biomarker potential of extracellular miRNAs, as it will be extremely difficult to distinguish between the origins of the extracellular miRNAs. Specifically, for myo-miRs, their (possibly low) disease specificity might be a cause for concern. Several studies have shown that extracellular myo-miRs are elevated in different muscular pathologies, including DMD and limb-girdle muscular dystrophies [11,66], which might indicate that elevated extracellular myo-miRs is a general marker of muscle pathology. Furthermore, the expression levels in the muscle are not always reflected in that found in serum or plasma. It has been shown that the increase in extracellular myo-miRs in serum and plasma is, in part, due to selective release of certain miRNAs during muscle growth and regeneration, instead of solely a passive leakage from damaged muscle and, therefore, more likely a complex function of the regenerative/degenerative status of the muscle, overall muscle mass, and tissue expression levels, which can have implications for their biomarker potential [67,68].

## 5. Therapeutic Potential

The dysregulation patterns of miRNAs found in DM1 and their links to the clinical phenotype also show their potential as therapeutics. The search for a therapeutic, targeting miRNAs in DM1, has focused mainly on miRNAs associated with the two splice factors involved in DM1 pathology, namely MBNL1 and CELF1 (Figure 2).

Four different miRNAs (miR-1, miR-30-5p, miR-23b, and miR-218) associated with the splice factor MBNL1 have been studied as therapeutic targets. As previously mentioned, pre-miR-1 processing is regulated by MBNL1. Sequestration of MBNL1 by toxic RNA results in downregulation of miR-1 and subsequent upregulation of downstream targets of miR-1. This, in turn, is linked to several DM1 symptoms, for example, one of the miR-1 targets is CACNA1C, which encodes for calcium channels in the heart. Misregulation may contribute to arrhythmias observed in DM1 patients [55]. However, no steps toward upregulating miR-1 to assess its therapeutic potential have been taken. miR-30-5p is another miRNA associated with the MBNL1 family, since it is a direct repressor of MBNL1-3 translation. A study showed that introducing a miR-30-5p mimic into C2C12 muscle cells downregulated the expression of MBNL and deregulated downstream targets of MBNL, including Trim55 (involved in sarcomere assembly) and Insulin Receptor (binding of insulin to the receptor initiates a downstream pathway that regulates muscle development) [69]. This shows an interesting possibility of using anti-miRs to repress miR-30-5p expression, increase MBNL protein levels, and alleviate associated DM1 symptoms. The last two miRNAs linked to MBNL1 were found by Cerro-Herreros et al. in 2018 [70]. miR-23b and miR-218 were identified as repressors of MBNL1 translation. Anti-miR treatment of each miRNA independently in DM1 myoblasts and DM1 mice increased MBNL1 levels, and in the latter this was linked to improvements in spliceopathy profile, histopathological signs, and functional myotonia without toxicity [70]. More recently, the same group has conducted a more in-depth study on the therapeutic potential of an anti-miR against miR-23b. They showed that the therapeutic effect is dose-dependent and has long-lasting effects. Subcutaneous administration of anti-miR 23b in human skeletal actin long repeat mice upregulated the expression of MBNL1 and rescued splicing alterations, grip strength, and myotonia [71].

For the splice factor CELF1, the focus has been primarily on miR-206, which may modulate CELF1 overexpression. The introduction of miR-206 mimics into cells overexpressing CEFL1 inhibited CELF1 expression and improved the myoblast fusion index and myotube area [72]. Another pair of miRNAs of note are miR-23a/b; they have shown to be downregulated by introducing expanded repeats, which leads to the upregulation of CELF1. This is another potential therapeutic ability worth exploring [56].

A recent study has discovered a new potential therapeutic target, namely miR-7. miR-7 was previously described to be downregulated in a DM1 *Drosophila* model and in muscle biopsies from patients [52]. In the current study, overexpression of miR-7 resulted in the rescue of DM1 myoblast fusion defects and myotube growth by repressing autophagy and the ubiquitin-proteasome system [73]. This improvement was independent of MBNL1. These results provide evidence for a new therapeutic candidate for muscle dysfunction in DM1.

## 6. Conclusions and Future Perspectives

miRNAs have great potential to become diagnostic tools in several diseases, including cancer, neurodegenerative diseases, and neuromuscular diseases. However, to date, insufficient knowledge and a lack of conclusive results to clarify the role of miRNA in disease diagnosis have held back the implementation of miRNA biomarkers in clinical settings [4]. In the past two decades, extensive research has been conducted in the miRNA expression profiles of DM1 patients and their biomarker potential. Several expression profiles have been found to be able to distinguish between DM1 patients and healthy subjects, and even a link has been made between progressive and non-progressive muscle wasting in DM1 patients. However, discrepancy between studies still exists. One of the main causes of the discrepancies seems to be the normalization process necessary for qPCR, for which no gold standard exists. A way around this might be the use of a fairly novel technique, the digital droplet PCR, which produces an absolute quantity, eliminating the need for normalization [74]. This will help in the comparability of results between groups. Further research, especially longitudinal studies, are needed to unravel the true biomarker potential of miRNAs in DM1 to see whether they can help in the prediction of disease progression and/or in the prediction of treatment efficacy.

## Figures and Tables

**Figure 1 jcm-09-03939-f001:**
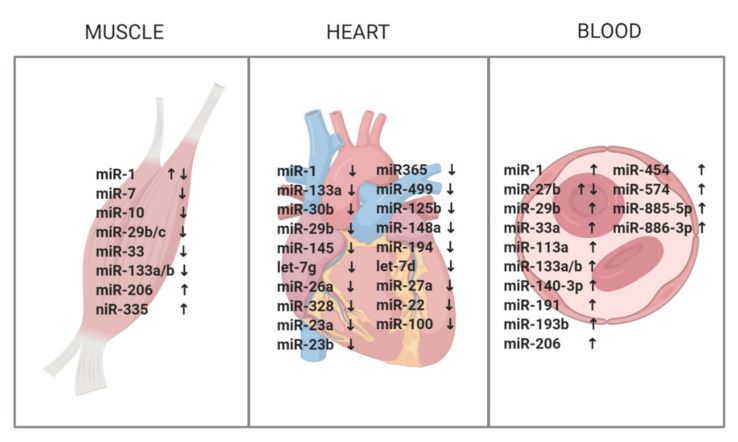
A summary of the studied miRNAs in skeletal muscle, heart, and blood. ↑ = upregulation, ↓ = downregulation. Created with BioRender.com.

**Figure 2 jcm-09-03939-f002:**
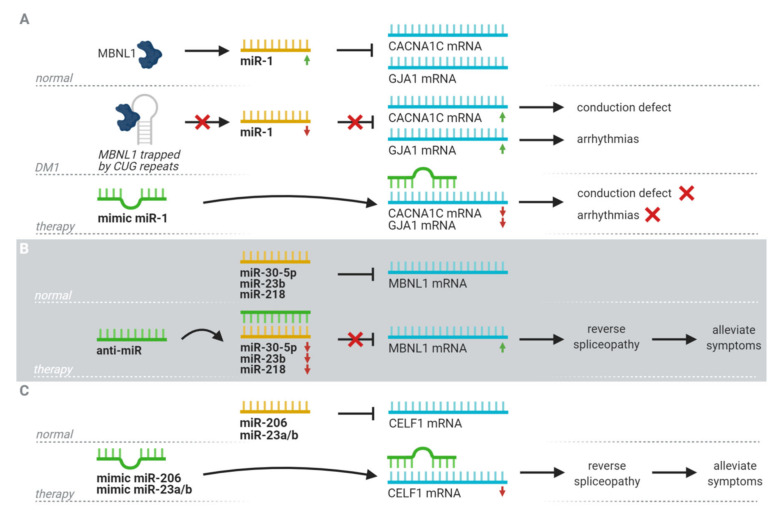
An overview of the suggested therapeutic potential of miRNAs in Myotonic Dystrophy type I (DM1) by targeting downstream targets of MBNL1, upregulating mRNA levels of MBNL1, or downregulating mRNA levels of CELF1 by the use of miRNA mimics or anti-miRs. (**A**) The use of an miR-1 mimic to circumvent the detrimental effect of MBNL1 sequestration, resulting in a downregulation of downstream targets and symptom alleviation. (**B**) The use of an anti-miR for miRNA inhibitors of MBNL1 to increase expression levels in DM1 and alleviate symptoms. (**C**) The use of a miRNA mimic of inhibitors of CELF1 to reduce its expression and alleviate symptoms in DM1. MBNL1 = muscleblind-like 1, CELF1 = CUG-BP and ETR-3-like factors family member 1, CACNA1C = Voltage-dependent L-type calcium channel subunit alpha-1C, GJA1 = connexin 43. Created with BioRender.com.

**Table 1 jcm-09-03939-t001:** Overview of the differentially expressed miRNAs analyzed in muscle biopsies.

Study	miRNAs		Muscle Biopsy	Sample Size	Normalization	Targets	Associated Pathological Signs and Mechanisms
Gambardella 2010 [48]	miR-206	↑	Vastus lateralis	7 DM1 4 ctrls	Hsa-let-7a		
Perbellini 2011 [51]	miR-1 * miR-29b/c ** miR-335 miR-33	↑ ↓ ↑ ↑	Biceps branchi	15 DM1 14 ctrls	miR-16	MEF2a *^,1^ MET *^,1^ GATA6 *^,1^ HAND2 *^,1^	muscle development ^1^
						KCNE1 *^,2^ KCNJ2 *^,2^ CALM2 *^,2^	Arrhythmia ^2^
						SFRS9 *^,3^	Splicing ^3^
						DAG1 **^,4^ DIABLO **^,4^ RET **TRIM63 **^,4^ TGFB3 **^,4^	Atrophy ^4^
Fernandez-Costa 2012 [52]	miR-1 *** miR-7 **** miR-10 *****	↓ ↓ ↓	Vastus lateralis Biceps branchi Deltoid	5 DM1 3 ctrls	Sno-RNA RNU48	SOD1 ***^,5^ SMARCA4 ***^,6^ NET 1 ***^,7^ CTSB ****^,8^ ATG4 ****^,9^ VCL ****^,10^ UBE11 *****^,11^	Free radical removal ^5^ Transcription ^6^ Apoptosis and signal transduction ^7^ Proteolysis ^8^ Autophagy ^9^ Cytoskeleton ^10^ Protein localization/activity regulation ^11^
Fritegotto 2017 [53]	miR-1 miR-133a/b miR206	↓ ↓ ↑	Vastus lateralis	12 DM1 6 ctrls	Hsa-let-7a		
Ambrose 2017 [52]	miR-1 miR-133a miR-29c	↓ ↓ ↓	Biceps branchi Gastrocnemius Deltoid	9 DM1 9 ctrls	Hsa-let-7a		

↑ = upregulation, ↓ = downregulation, DM1 = DM1 patients, ctrls = healthy subjects, colors indicate to which miRNA the targets belong. Normalization stands for the miRNA, either endogenous or spike-in, that has been used to normalize the differentially expressed miRNAs. *, **, ***, ****, *****, indicate to which miRNA the targets belong. ^1–11^ numbers indicates which Associated Pathological Signs and Mechanisms belong to which target.

**Table 2 jcm-09-03939-t002:** Overview of the differentially expressed miRNAs analyzed in heart biopsies.

Study	miRNAs		Heart	Sample Size	Normalization	Targets	Associated Pathological Signs
Rau 2011 [55]	miR-1	↓	Left ventricle	5 DM1 8 ctrls	U6 snRNA	CACNA1C ^1^ GJA1 ^2^	Arrhythmias ^1^ Conduction ^2^
Kalsotra 2014 [56]	miR-1 * miR-133 ** miR-30b ** miR-29b ** miR-145 let-7g miR-26a miR-328 miR-23a miR-23b miR365 miR-499 miR-125b miR-148a miR-194 let-7d miR-27a miR-22 miR-100	↓ ↓ ↓ ↓ ↓ ↓ ↓ ↓ ↓ ↓ ↓ ↓ ↓ ↓ ↓ ↓ ↓ ↓ ↓	unknown	8 DM1 4 ctrls	MammU6		Conduction * Fibrosis **

↑ = upregulation, ↓ = downregulation, DM1 = DM1 patients, ctrls = healthy subjects, colors indicate to which miRNA the proposed disease involvement belongs. Normalization stands for the miRNA, either endogenous or spike-in, that has been used to normalize the differentially expressed miRNAs. *, ** indicate to which miRNA the associated pathological signs belong. ^1,2^ numbers indicates which Associated Pathological Signs belong to which target.

**Table 3 jcm-09-03939-t003:** Overview of the differentially expressed miRNAs analyzed in blood.

Study	miRNAs		Blood	Sample Size	Normalization	Associated Clinical Characteristics
Perfetti 2014 [33]	miR-113a miR-193b miR-191 miR-454 miR-574 miR-885-5p miR-886-3p miR-27b	↑ ↑ ↑ ↑ ↑ ↑ ↑ ↓	Plasma	36 DM1 36 ctrls	Cel-miR-39 miR-17-5p miR-106a	Negative correlation to MRC score Positive correlation to MIRS scale
Koutsoulidou 2015 [34]	miR-1 miR-133a/b miR-206	↑ ↑ ↑	Serum	23 DM1 23 ctrls	miR-16	Increased levels in progressive muscle wasting compared to stable muscle wasting
Perfetti 2016 [58]	miR-1 miR-133a/b miR-206 miR-140-3p miR-454 miR-574 miR-27b	↑ ↑ ↑ ↑ ↑ ↑ ↓	Plasma	103 DM1 111 ctrls	Cel-miR-39 miR-17-5p miR-106a	Negative correlation to muscle strength Non-significant correlation to CK levels
Koutsoulidou 2017 [59]	miR-1 miR-133a/b miR-206 miR-113a miR-193b miR-191 miR-574 miR-885-5p miR-886-3p miR-27b	↑ ↑ ↑ ↑ ↑ ↑ ↑ ↑ ↑ ↑	Serum	63 DM1 63 ctrls	miR-16	Increased levels in progressive muscle wasting compared to stable muscle wasting
Pegoraro 2020 [51]	miR-1 miR-133a/b miR-206	↑ ↑* ↑	Serum	9 DM1 7 ctrls	miR-39-3p C. elegans	After a six-week exercise training the 4 miRNAs significantly decreased in parallel with improved muscle function
Ambrose 2017 [52]	miR-133a miR-29b miR-33a	↑ ↑ ↑	Whole blood	10 DM1 10 ctrls	Hsa-miR-183	

↑ = upregulation, ↓ = downregulation, DM1 = DM1 patients, ctrls = healthy subjects, * only in a subset of patients. Normalization stands for the miRNA, either endogenous or spike-in, that has been used to normalize the differentially expressed miRNAs. General associated clinical characteristics are indicated per study.

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
