# Peer review of "The Biomarker Potential of miRNAs in Myotonic Dystrophy Type I"

_jcm, 2020, doi:10.3390/jcm9123939_

Round 1

Reviewer 1 Report

Koehorst et al. present a review of miRNA biomarkers in the context of DM1. This is an interesting topic and I learned some interesting things from reading it. However, I feel that this article does not really engage with the literature (particularly in terms of uncritical restatement of findings from a few low quality studies, examples below), and that it lacks some insight.

Major Comments

  1. The authors state that:

‘In DM1, the need for a biomarker that can predict disease progression is essential, since a treatment to cure the disease or at least halt progression is not yet developed.’

I am not sure that this completely follows. The authors should expand on the rationale for using miRNA biomarkers for DM1. Why would having biomarkers help in the context of having no therapies? How would the miRNA biomarkers be used to manage disease or affect clinical decisions?

  1. Following on from the last point, there is no discussion of current biomarker strategies. Do extracellular miRNA biomarkers offer any advantage over serum CK?
  2. The authors state that:

‘miR-206, the latter only expressed in skeletal muscle’

This is not strictly true (although often written). miR-206 can be detected in multiple tissues, including cardiac muscle. However, this is at much lower levels than observed for the other myomiRs.

In general, the authors should be careful about making strongly worded statements regarding miRNA expression. It is probably more accurate to say that a miRNA is enriched in a given tissue:

e.g. miR-1, miR-133 and miR-206 are highly enriched in skeletal muscle, miR-1 and miR-133 are also highly enriched in cardiac muscle.

  1. The authors state:

‘a computational analysis revealed that miR-103 and miR-107 were 131 binding preferentially to mutated DMPK mRNA’

Do the authors mean that binding to DMPK was predicted? Was there any experimental evidence for this?

  1. The authors state:

‘Only miR-206 was found to be overexpressed in biopsies of DM1 patients. However, the predicted target genes of miR-206, such as Utrophin, were not altered, questioning its functional relevance’.

I think this is a very simplistic interpretation of these findings. Muscle tissue is composed of multiple types of myofibres, and mononuclear cells. Gene expression analyses in bulk tissue may miss some of the subtlety of gene regulatory events as differences may be averaged across cell types. For example, miR-206 is known to be highly-enriched in regenerating fibres (Yuasa et al. 2008). Hypothetically, the emergence of regenerating fibres in the disease state manifests as a pronounced increase in miR-206 in bulk tissue, even if regenerating fibres constitute only a small minority of total fibres. Utrophin (assuming it is a bona fide miR-206 target) may therefore be downregulated in regenerating fibres. However, measuring utrophin in bulk tissue, may reveal no change, as expression of this protein is not changed in most fibre types.

As an aside, I think it would be very helpful and interesting to include discussion of DM1 histopathology. My understanding is that DM1 muscle does exhibit centrally nucleated fibres, but I am not sure that these are truly regenerating. Perhaps the authors could clarify, discuss this point?  I feel it is very important, as regenerative pathology is a prominent explanation for extracellular myomiR release in the unrelated muscular dystrophy, DMD.

  1. In general, the authors should discuss DM1 miRNA biomarkers in the context of other diseases of muscle. Extracellular myomiRs are elevated in many conditions, and may be general markers of muscle pathology.

https://pubmed.ncbi.nlm.nih.gov/23418438/

Furthermore, work in other contexts has provided deeper insights into the reasons for elevated miRNAs in the circulation:

https://pubmed.ncbi.nlm.nih.gov/27466195/

https://pubmed.ncbi.nlm.nih.gov/30219269/

  1. At several places in the manuscript, the authors discuss differences between miR-133a and miR-133b, or how the measurement of these miRNAs can be combined for some purpose. I am very sceptical of these inferences. These two miRNAs differ in only their 3prime terminal nucleotide. This minor difference cannot be distinguished by stem loop RT small RNA TaqMan (Thermo Fisher) or poly A tailing (miScript, Qiagen) technologies.
  2. The authors reference a paper by Koutsoulidou et al. (2017) that reports that myomiRs are primarily found in exosomes. Importantly, this study used a polymer precipitation method to prepare exosomes. These methods precipitate macromolecules in general and are not specific to exosomes.

From the ISEV position paper on EV isolation: https://www.ncbi.nlm.nih.gov/pmc/articles/PMC4275645/

‘In particular, polymer-based methods to precipitate EVs (used by some commercial kits) do not exclusively isolate EVs, and are likely to co-isolate other molecules, including RNA–protein complexes’.

The situation is even worse for miRNAs, as these kits preferentially precipitate non-exosomal miRNAs: https://pubmed.ncbi.nlm.nih.gov/30574280/

Studies which report exosome/EV results and rely only on these methods should be treated with much caution.

As an aside, the term ‘exosomes’ should be avoided. The techniques used are unable to distinguish exosomes from other vesicles with different biosynthetic origins but similar biophysical properties (e.g. microvesicles). The term Extracellular Vesicles (EVs) should be used instead.

9. Differences between normalization methods is presented as a possible reason for discrepancies between DM1 miRNA. Indeed, total miRNA content has been shown to be a dependent variable in other contexts (and so normalization to global Cq value may introduce additional error):

https://pubmed.ncbi.nlm.nih.gov/24586621/

Minor Comments

  1. The ‘±’ symbol should probably be replaced with ‘~’.
  2. In the sentence ‘MicroRNAs (miRNAs) are small, single-stranded, ±22 nucleotide-long RNAs that control regulation,’ ‘control regulation’ is unusual wording, and unclear.
  3. Line 43: change ‘30 UTRs’ to 3ʹ UTR’
  4. Possible erroneous paragraph break at line 198
  5. Line 256, I believe ‘miR-113a’ should be ‘miR-133a’
  6. Line 367, I think ‘miR-3-5p’ should be ‘miR-30-5p’

Author Response

Reviewer 1

Comments and Suggestions for Authors

Koehorst et al. present a review of miRNA biomarkers in the context of DM1. This is an interesting topic and I learned some interesting things from reading it. However, I feel that this article does not really engage with the literature (particularly in terms of uncritical restatement of findings from a few low quality studies, examples below), and that it lacks some insight.

Major Comments

  1. The authors state that:

‘In DM1, the need for a biomarker that can predict disease progression is essential, since a treatment to cure the disease or at least halt progression is not yet developed.’

I am not sure that this completely follows. The authors should expand on the rationale for using miRNA biomarkers for DM1. Why would having biomarkers help in the context of having no therapies? How would the miRNA biomarkers be used to manage disease or affect clinical decisions?

Thank you for your comment. We have elaborated on the sentence with a small discussion on the rational of needing biomarkers in DM1 in line 102 through 124:

In DM1, the need for a biomarker that can predict disease outcomes is essential, especially for cardiac and respiratory complications, to improve patient’s survival. Symptoms and disease progression are highly variable among patients, and life expectancy is significantly decreased, primarily due to respiratory failure and cardiac events [28]. This high variability and the multi-systemic nature of the disease present a unique challenge in terms of disease management. Currently, disease management is mostly focused on symptomatic treatment and preserving function and independence. Current assessment strategies include genetic testing, electromyography, skeletal muscle histopathology and magnetic resonance. The only circulating marker used to date is creatine kinase, but this is not elevated in all patients and lacks disease specificity, since it is altered in most disorders involving skeletal muscle damage [29], as well as intense physical activity [30]. Another suggested biomarker are the alternative splicing changes observed in the DM1 skeletal muscle [31]. However, monitoring them would require several invasive muscle biopsies. miRNAs have the potential to help estimate the degree and type of involvement of symptoms before they appear, and to apply the appropriate preventive measures, for example several studies have indicated that miRNAs correlate to muscle strength and disease stage in DM1 [32, 33]. In addition to disease management, miRNAs could be used as biomarkers for future clinical trials. Currently, outcome measures consist mostly of clinical and functional ones, such as cognitive function assessment by questionnaires, patient-reported outcomes such as DM1-activ, muscle testing by the 6-minute walk test, grip strength, MIRS scale and myotonia degree [34]. These can be, however, influenced by several confounding factors. For example, the 6-minute walk test is not only influenced by muscle strength, but also by factors such as the level of fatigue and motivation [35]. Also, these tests might lack sensitivity to subtle changes happening in the rather short time frame of most clinical trials.     

  1. Following on from the last point, there is no discussion of current biomarker strategies. Do extracellular miRNA biomarkers offer any advantage over serum CK?

Thank you for your comment. Serum CK is a general marker for skeletal muscle damage, and is not altered in all DM1 patients. According to the data in the literature, it is reasonable to hypothezised that miRNA have the potential to give us more information on symptomatology onset, severity and progression, although much work is still to be done to fully elucidate the role of miRNAs in skeletal muscle, cardiac and overall function of the human body. Within the previous discussion, we have added the current biomarker strategies, which as you will see are, apart from serum CK, non-existent. Alternative splicing has been proposed as a potential biomarker, however the invasive procedure (muscle biopsy) needed to monitor alternative splicing will hamper its use in the clinical setting. This can be found in line 109 through 114:

The only circulating marker used to date is creatine kinase, but this is not elevated in all patients and lacks disease specificity, since it is altered in most disorders involving skeletal muscle damage [29], as well as intense physical activity [30]. Another suggested biomarker are the alternative splicing changes observed in the DM1 skeletal muscle [31]. However, monitoring them would require several invasive muscle biopsies.

  1. The authors state that:

‘miR-206, the latter only expressed in skeletal muscle’

This is not strictly true (although often written). miR-206 can be detected in multiple tissues, including cardiac muscle. However, this is at much lower levels than observed for the other myomiRs.

In general, the authors should be careful about making strongly worded statements regarding miRNA expression. It is probably more accurate to say that a miRNA is enriched in a given tissue:

e.g. miR-1, miR-133 and miR-206 are highly enriched in skeletal muscle, miR-1 and miR-133 are also highly enriched in cardiac muscle.

Thank you for your comment and our apology for the wrong statement. We have corrected the sentence in line 132 – line 134.

‘They consist of miR-1, miR-133a/b, and miR-206, which are highly enriched in skeletal muscle, whereas miR-1 and miR-133a/b are also highly enriched in cardiac muscle.’

  1. The authors state:

‘a computational analysis revealed that miR-103 and miR-107 were 131 binding preferentially to mutated DMPK mRNA’

Do the authors mean that binding to DMPK was predicted? Was there any experimental evidence for this?

Thank you for your comment. Yes, we meant that binding to DMPK mRNA was predicted by a computational analysis. To clarify this point and how it was predicted, we have reshaped the sentence and added some additional information, which is in line 151 through 158:

‘They focused on the myo-miRs, in addition to miR-103 and miR-107, which were proposed to be attractive candidates for binding DMPK mRNA [47]. More specifically, a computational analysis revealed that miR-103 and miR-107 would repress the DMPK wild-type allele with 15% and that repression of mRNA by CTG repeat-binding miRNAs would increase with the number of CTGs in a their 3’UTR. They proposed a model in which CTG repeat-binding miRNAs, such as miR107 and miR-103, preferentially bind to mutated DMPK mRNA, which could have a miRNA-leaching effect and therefore implications for DM1 pathology [47].’

At the moment, we are not aware of any experimental evidence for this computational analysis.

  1. The authors state:

‘Only miR-206 was found to be overexpressed in biopsies of DM1 patients. However, the predicted target genes of miR-206, such as Utrophin, were not altered, questioning its functional relevance’.

I think this is a very simplistic interpretation of these findings. Muscle tissue is composed of multiple types of myofibres, and mononuclear cells. Gene expression analyses in bulk tissue may miss some of the subtlety of gene regulatory events as differences may be averaged across cell types. For example, miR-206 is known to be highly-enriched in regenerating fibres (Yuasa et al. 2008). Hypothetically, the emergence of regenerating fibres in the disease state manifests as a pronounced increase in miR-206 in bulk tissue, even if regenerating fibres constitute only a small minority of total fibres. Utrophin (assuming it is a bona fide miR-206 target) may therefore be downregulated in regenerating fibres. However, measuring utrophin in bulk tissue, may reveal no change, as expression of this protein is not changed in most fibre types.

Thank you for your comment and the information provided. We agree that our interpretation lacked finesse and we have decided to remove this comment from the manuscript and have stated the finding and the possible explanation you so generously have provided. The changes can be found in line 158 – line 164

‘Only miR-206 was found to be overexpressed in biopsies of DM1 patients. However, the predicted target genes of miR-206, such as Utrophin, were not altered. A potential explanation can be the presence of multiple myofibers in skeletal muscle. For example, miR-206 is known to be highly enriched in regenerating fibers [48], which constitutes only a small portion of the bulk tissue measured. Although miR-206 shows a pronounced increase due to the addition of regenerating fibers, this does not mean that its target genes are equally affected, and the minor change might not be detected in bulk tissue.’

  1. As an aside, I think it would be very helpful and interesting to include discussion of DM1 histopathology. My understanding is that DM1 muscle does exhibit centrally nucleated fibres, but I am not sure that these are truly regenerating. Perhaps the authors could clarify, discuss this point? I feel it is very important, as regenerative pathology is a prominent explanation for extracellular myomiR release in the unrelated muscular dystrophy, DMD.

Thank you for the comment. It has made us search and update our knowledge on histopathology in DM1. First of all, we have to say that there are few histopathology studies made in DM1, because the diagnosis is well established through clinical and genetic determination, and muscle biopsy is not needed. Moreover, disease manifestation is mainly in distal muscles, but the studies are very diverse from the muscle group perspective, as either biceps, deltoids, quadriceps and/or vastus lateralis have been analysed. The studies are also diverse in the type of patients analysed, and since this disease can appear at any age, probably the same histological alterations would not apply to a congenital, infantile, juvenile, adult or late onset patient, while most of the studies include adult patients with no age distinctions. Having said that, we have found that some of the studies coincide in finding a selective muscle type I fibre atrophy and type I fibre predominance. Fibrosis, fat infiltration and necrotic fibres were rarely found. Central nuclei can be found, especially in atrophied type I fibres, but seems not to be a major finding among the muscle sections. Of course, there is no evidence on whether these few central nuclei are an effect of regeneration or due to other causes of this disease. We also searched for utrophin studies in DM1 but we were unable to find anything. So, we are sorry we cannot shed more light on the histological knowledge of DM1, but the data is so scarce and so diverse that we found that we did not have enough resources to implement this in our discussion.

https://pubmed.ncbi.nlm.nih.gov/23169601/

https://pubmed.ncbi.nlm.nih.gov/26708183/

https://pubmed.ncbi.nlm.nih.gov/8065390/

https://pubmed.ncbi.nlm.nih.gov/12796551/

https://pubmed.ncbi.nlm.nih.gov/21630234/

  1. In general, the authors should discuss DM1 miRNA biomarkers in the context of other diseases of muscle. Extracellular myomiRs are elevated in many conditions, and may be general markers of muscle pathology.

https://pubmed.ncbi.nlm.nih.gov/23418438/

Furthermore, work in other contexts has provided deeper insights into the reasons for elevated miRNAs in the circulation:

https://pubmed.ncbi.nlm.nih.gov/27466195/

https://pubmed.ncbi.nlm.nih.gov/30219269/

Thank you for your comment. We agree that a discussion on extracellular myo-miRs and its origin was lacking and we have added a discussion on the topic at the end of the serum and plasma section in line 388 through line 398:

‘Specifically for myo-miRs, their (possibly low) disease specificity might be a cause for concern. Several studies have shown that extracellular myo-miRs are elevated in different muscular pathologies, including DMD and limb-girdle muscular dystrophies [11, 64], which might indicate that elevated extracellular myo-miRs is a general marker of muscle pathology. Furthermore, the expression levels of the muscle are not always reflected in the ones found in serum or plasma. It has been shown that the increase of extracellular myo-miRs in serum and plasma is in part due to selective release of certain miRNAs during muscle growth and regeneration, instead of solely a passive leakage from damaged muscle and therefore more likely a complex function of the regenerative/degenerative status of the muscle, overall muscle mass, and tissue expression levels, which can have implications for their biomarker potential [65, 66].’

  1. At several places in the manuscript, the authors discuss differences between miR-133a and miR-133b, or how the measurement of these miRNAs can be combined for some purpose. I am very sceptical of these inferences. These two miRNAs differ in only their 3prime terminal nucleotide. This minor difference cannot be distinguished by stem loop RT small RNA TaqMan (Thermo Fisher) or poly A tailing (miScript, Qiagen) technologies.

Thank you for your comment. We understand your concern regarding the ability to differentiate successfully between miR133a and miR133b, based on the fact that the mature miRNA sequence only differs in one nucleotide at their 3’ end. However, stem loop primers are designed to be highly specific and contain in their sequence the last 6 nucleotides of the miRNA, which includes this one nucleotide of difference between miR133a and 133b. Some authors have evaluated whether stem loop primers are able to differentiate a single nucleotide difference. For example, a study showed that the differences between Let-7a/b/c/d/e, which ranges between one and five nucleotides was able to distinguish them with high specificity. More specifically, a one nucleotide difference resulted only in 0.1-3.7% of non-specific signal: https://pubmed.ncbi.nlm.nih.gov/16314309/. Furthermore, the differentiation between miR-133a and miR-133b seems to be common practice in both the field of DM1, as well as other diseases such as DMD: https://pubmed.ncbi.nlm.nih.gov/32350552/ and differences can be found in their expression levels, since some of the papers only see an upregulation in only one of the two miRNAs. So, the methodology seems robust and we do not have any reason to not trust the results obtained.

  1. The authors reference a paper by Koutsoulidou et al. (2017) that reports that myomiRs are primarily found in exosomes. Importantly, this study used a polymer precipitation method to prepare exosomes. These methods precipitate macromolecules in general and are not specific to exosomes.

From the ISEV position paper on EV isolation: https://www.ncbi.nlm.nih.gov/pmc/articles/PMC4275645/

In particular, polymer-based methods to precipitate EVs (used by some commercial kits) do not exclusively isolate EVs, and are likely to co-isolate other molecules, including RNA–protein complexes’.

The situation is even worse for miRNAs, as these kits preferentially precipitate non-exosomal miRNAs: https://pubmed.ncbi.nlm.nih.gov/30574280/

Studies which report exosome/EV results and rely only on these methods should be treated with much caution.

Thank you for your comments. We were unaware of this problem with EV isolation and have included a cautionary mention in our manuscript in line 332 – line 342:

‘A novel finding was the encapsulation of these myo-miRs in EVs: it was reported that the majority of the differentially expressed myo-miRs in DM1 patients could be found in EVs. The levels of the four myo-miRs were significantly higher in EVs isolated from serum of the DM1 patients compared to controls and seem to discriminate between progressive and stable DM1 patients. However, caution should be taken with regards to these results. It has been shown that polymer-based methods to precipitate EVs (such as the ExoquickTM Exosome Precipitation solution used in this study) are not exclusively precipitating EVs and co-isolation of other molecules, including RNA-Protein complexes is a possibility [58]. Especially in the case of miRNAs, it has been shown that these kits may preferentially precipitate non-exosomal miRNAs [59]. This means that the miRNAs found in this precipitate may not exclusively originate from EVs or in fact might primarily be non-exosomal.’

  1. As an aside, the term ‘exosomes’ should be avoided. The techniques used are unable to distinguish exosomes from other vesicles with different biosynthetic origins but similar biophysical properties (e.g. microvesicles). The term Extracellular Vesicles (EVs) should be used instead.

Thank you for this comment. We used the term that was used in the publication described and were unaware this was a term that was not suitable for these type of vesicles. We have changed the name to your suggestion, which can be found in line 332 – line 342.  

  1. Differences between normalization methods is presented as a possible reason for discrepancies between DM1 miRNA. Indeed, total miRNA content has been shown to be a dependent variable in other contexts (and so normalization to global Cq value may introduce additional error):

https://pubmed.ncbi.nlm.nih.gov/24586621/

Thank you for this addition to our discussion on normalization methods. We have implemented this publication in line 215 - 218

‘In addition, miRNA expression levels are studied by quantitative PCR (qPCR), where expression levels are normalized against an endogenous miRNA of which the levels are known to be stable or by using a spike-in miRNA. Total miRNA content has been shown to be an experimental variable and adds a systematic bias in miRNA quantification [52].’

Minor Comments

The ‘±’ symbol should probably be replaced with ‘~’.

The symbol has been replaced in line 33

In the sentence ‘MicroRNAs (miRNAs) are small, single-stranded, ±22 nucleotide-long RNAs that control regulation,’ ‘control regulation’ is unusual wording, and unclear.

Thank you for your comment. ‘Control regulation’ has been changed to ‘regulate gene-expression’ in line 33 and 34.

Line 43: change ‘30 UTRs’ to 3ʹ UTR’

Thank you for this observation, the sentence has been corrected in line 43

Possible erroneous paragraph break at line 198

Thank you for this observation, you were absolutely right. The erroneous break has been removed at line 230

Line 256, I believe ‘miR-113a’ should be ‘miR-133a’

Thank you for this observation, the mistake has been corrected in line 287

Line 367, I think ‘miR-3-5p’ should be ‘miR-30-5p’

Thank you for this observation, the mistake has been corrected in line 425

On behalf of all authors, many thanks for this insightful review.

Reviewer 2 Report

The review by Koehorst and col. entitled "The biomarker potential of miRNAs in myotonic dystrophy type 1" outlines the current status of miRNAs on their potential use as biomarkers in myotonic dystrophy type 1. Despite a field in a very early stage of development, with information still dispersed through different research groups' work, the authors have done good work on compiling and presenting the data.

The manuscript only requires minor revision in specific parts of the review to achieve an improved flow for the information provided. Specifically, it is suggested:

  1. The miRNAs' usefulness as biomarker candidates is mainly presented through all the manuscript based on their detection in bodily fluids. However, why miRNAs could be considered as ideal biomarkers, compared to another type of molecules, should include some text on additional biological/technical features as specificity and sensitivity, but also some discussion on other key aspects important for their potential development as their potential translatability to the clinic, or possibility for cheap and low time-consuming use.
  2. On the line of the first comment, how circulating miRNAs reach the fluids is an important point for evaluation and/or discussion. Exosomes are not the only (and not the primary source) way for it. Also interesting to discuss which are the tissue sources providing miRNA to fluids in DM1. Is it possible to elucidate it when working with a multi-systemic disease like DM1 that could be sending miRNAs from different sources? Something should be mentioned in this matter through the text.
  3. What type of biomarkers are currently used for clinical trials in DM1, and why it is needed to get better ones should be somehow included through the text.
  4. The sentence "miRNAs have the potential to help estimate the degree and type of involvement of symptoms before they appear and to apply the appropriate preventative measures" (lines 100-102) need specific examples/references.
  5. New data on miRNAs and myotonic dystrophy have been recently published that are directly connected to data presented through the text for miR-23b and miR-7 (Cerro-Herrero et al., 2020; and Sabater-Arcís et al., 2020).

Author Response

Reviewer 2

Comments and Suggestions for Authors

The review by Koehorst and col. entitled "The biomarker potential of miRNAs in myotonic dystrophy type 1" outlines the current status of miRNAs on their potential use as biomarkers in myotonic dystrophy type 1. Despite a field in a very early stage of development, with information still dispersed through different research groups' work, the authors have done good work on compiling and presenting the data.

The manuscript only requires minor revision in specific parts of the review to achieve an improved flow for the information provided. Specifically, it is suggested:

  1. The miRNAs' usefulness as biomarker candidates is mainly presented through all the manuscript based on their detection in bodily fluids. However, why miRNAs could be considered as ideal biomarkers, compared to another type of molecules, should include some text on additional biological/technical features as specificity and sensitivity, but also some discussion on other key aspects important for their potential development as their potential translatability to the clinic, or possibility for cheap and low time-consuming use.

Thank you for your comment. We agree that there are more advantages to the use of miRNAs as biomarkers and have added additional benefits to the manuscript in line 49 – 53:

‘This makes them excellent biomarker candidates, since one of the most important properties of a good biomarker is easy accessibility. Furthermore, their detection is easily achieved with low-cost techniques, such as RT-qPCR. In addition, the discovery that miRNA expression profiles are different in diseased state and their tissue-specific expression patterns, further strengthened their biomarker potential and extensive research has been conducted into this.’

  1. On the line of the first comment, how circulating miRNAs reach the fluids is an important point for evaluation and/or discussion. Exosomes are not the only (and not the primary source) way for it. Also interesting to discuss which are the tissue sources providing miRNA to fluids in DM1. Is it possible to elucidate it when working with a multi-systemic disease like DM1 that could be sending miRNAs from different sources? Something should be mentioned in this matter through the text.

Thank you for your comment. We agree that a mentioning of how on circulating miRNAs enter the bloodstreams was lacking from the manuscript and we have added this in line 46 - 49

‘There are several ways in which miRNAs can reach the bloodstream from the tissues, including lipid or lipoprotein complexes and extracellular vesicles (EVs), but they have also been found as free-floating complexes with AGO proteins [5].’

In addition, we have added a small discussion in the serum and plasma section of the manuscript on the multi-systemic nature of the disease and the difficulty it will present on elucidating the origin of the miRNA in line 384 – line 390:

‘Although some miRNAs are tissue-specific, there is still a major overlap between the expression of miRNAs. For example, although miRNA-206 is highly enriched in skeletal muscle, it can also be detected in cardiac muscle and its origin might be from both. The same holds true for miR-1 and miR-133, which are highly enriched in both tissues. The multi-systemic nature of DM1 may add additional complexity to determining the biomarker potential of extracellular miRNAs, as it will be extremely difficult to distinguish between the origins of the extracellular miRNAs.’

  1. What type of biomarkers are currently used for clinical trials in DM1, and why it is needed to get better ones should be somehow included through the text.

Thank you for your comment. Biomarkers in clinical trials is an interesting topic and currently no biomarkers are available. The outcome measures depend mostly on clinical and functional outcomes, such as cognitive tests, the 6-minute walking test and muscle weakness scales, such as the MIRS scale. These outcomes are heavily influenced by other factors such as motivation and for example the level of fatique or the presence of day time sleepiness. In addition, the changes in these outcomes might be too subtle in the timeframe of a clinical trial. A discussion on the topic has been added in line 117 – line 124:

In addition to disease management, miRNAs could be used as biomarkers for future clinical trials. Currently, outcome measures consist mostly of clinical and functional ones, such as cognitive function assessment by questionnaires, patient-reported outcomes such as DM1-activ, muscle testing by the 6-minute walk test, grip strength, MIRS scale and myotonia degree [34]. These can be, however, influenced by several confounding factors. For example, the 6-minute walk test is not only influenced by muscle strength, but also by factors such as the level of fatigue and motivation [35]. Also, these tests might lack sensitivity to subtle changes happening in the rather short time frame of most clinical trials.      ‘   

  1. The sentence "miRNAs have the potential to help estimate the degree and type of involvement of symptoms before they appear and to apply the appropriate preventative measures" (lines 100-102) need specific examples/references.

Thank you for your comment. We agree that we might have been to concise in our wording here and have elaborated on the point we were trying to make. This can be found in line 102 through line 117:

In DM1, the need for a biomarker that can predict disease outcomes is essential, especially for cardiac and respiratory complications, to improve patient’s survival. Symptoms and disease progression are highly variable among patients, and life expectancy is significantly decreased, primarily due to respiratory failure and cardiac events [28]. This high variability and the multi-systemic nature of the disease present a unique challenge in terms of disease management. Currently, disease management is mostly focused on symptomatic treatment and preserving function and independence. Current assessment strategies include genetic testing, electromyography, skeletal muscle histopathology and magnetic resonance. The only circulating marker used to date is creatine kinase, but this is not elevated in all patients and lacks disease specificity, since it is altered in most disorders involving skeletal muscle damage [29], as well as intense physical activity [30]. Another suggested biomarker are the alternative splicing changes observed in the DM1 skeletal muscle [31]. However, monitoring them would require several invasive muscle biopsies. miRNAs have the potential to help estimate the degree and type of involvement of symptoms before they appear, and to apply the appropriate preventive measures, for example several studies have indicated that miRNAs correlate to muscle strength and disease stage in DM1 [32, 33].

  1. New data on miRNAs and myotonic dystrophy have been recently published that are directly connected to data presented through the text for miR-23b and miR-7 (Cerro-Herrero et al., 2020; and Sabater-Arcís et al., 2020).

Thank you for this information and our apologies that we missed these new publications. We have added them to the section Therapeutic potential in line 430 – line 434:

‘More recently, the same group has conducted a more in-depth study on the therapeutic potential of an anti-miR against miR-23b. They showed that the therapeutic effect is dose-dependent and has long-lasting effects. Subcutaneous administration of the anti-miR 23b in human skeletal actin long repeat mice upregulated the expression of MBNL1 and rescued splicing alterations, grip strength and myotonia [69].’

And line 441 through line 446

‘A recent study has discovered a new potential therapeutic target, namely miR-7. miR-7 was previously described to be downregulated in a DM1 drosophila model and in muscle biopsies from patients [50]. In the current study, overexpression of miR-7 resulted in the rescue of DM1 myoblast fusion defects and myotube growth by repressing autophagy and the ubiquitin-proteasome system [71]. This improvement was independent of MBNL1. These results provide evidence for a new therapeutic candidate for muscle dysfunction in DM1.